

# Myths in psychology: psychological misconceptions among Spanish psychology students

Cristina Rodríguez-Prada[1], Cristina Orgaz[2] and Carmelo P. Cubillas[1]

[1] Department of Experimental Psychology, Universidad Autónoma de Madrid, Madrid, Spain
[2] Department of Experimental Psychology, Universidad Nacional de Educación a Distancia, Madrid, Spain

## ABSTRACT

Myths in Psychology are beliefs that are widely spread and inconsistent with the empirical evidence available within this field of knowledge. They are characterized by being relatively stable, resistant to change, and prevalent both among the non-academic population and among students and professionals within this discipline. The aim of this study was to analyse the prevalence of these myths among Spanish psychology students and the influence of three variables: the type of university, face-to-face (UAM) and online (UNED), the academic year in which participants were enrolled and familiarity with scientific dissemination. Results show that participants from the face-to-face university, enrolled in higher academic years and that reports familiarity with scientific dissemination believe less in myths than those from the online university, enrolled in lower years and that report no familiarity with scientific dissemination.

## INTRODUCTION

Myths in Psychology are defined as beliefs about human behavior that are inconsistent with the available scientific evidence but that exhibit great social support (*Lilienfeld et al., 2010*; *Stanovich, 1992*). Contrary to certain expectations, these misconceptions are not only believed by people outside of Psychology, but also prevail among professionals within this discipline (*Fasce & Adrián-Ventura, 2020*; *Furnham & Hughes, 2014*; *Galassi & Gersh, 1993*; *Hooper, 2006*; *Houben et al., 2019*; *Lilienfeld et al., 2013*; *Torres, Boccacini & Miller, 2006*). Although some studies show that university studies reduce the belief in myths (*Bensley, Lilienfeld & Powell, 2014*; *Hughes et al., 2015*; *Sibicky, Klein & Embrescia, 2020*), it has also been shown that having university studies does not eliminate the belief in myths nor are these beliefs fully replaced with explanations from scientific psychology (*Hughes et al., 2015*; *Lyddy & Hughes, 2012*; *Root & Stanley, 2017*). On many occasions, students who begin their studies in Psychology have imprecise and contradictory information about human behavior (*Hughes, Lyddy & Lambe, 2013*). The resistance to change that characterizes these ideas is due to a history of intermittent and repeated exposure to these false facts through socializing agents such as the media, family or peers (*Furnham & Hughes, 2014*; *Kowalski & Taylor, 2004*). This favors the creation of a confirmation bias when presented with

Corresponding author
Carmelo P. Cubillas,
carmelo.perez@uam.es

new information (*Furnham & Hughes, 2014*; *Kowalski & Taylor, 2004*). In this way, most popular knowledge is not disproven and cannot be replaced by more adjusted beliefs (*Lilienfeld et al., 2010*; *Stanovich, 1992*), which leads, in many cases, to its consequences reaching the professional practice of those students who complete this stage of education (*Furnham & Hughes, 2014*).

Believing that some erroneous statements in psychology are true is not a trivial phenomenon and can lead to serious negative effects, such as the implementation of harmful practices in educational or clinical areas, for example (*Lilienfeld et al., 2010*). The belief in myths also hinders the development of learning content related to the discipline and critical thinking within and outside of Psychology (*Lilienfeld et al., 2010*). It can even legitimize existing inequalities in society, such as the differences between men and women, the stigma towards people with mental health problems, or the consequences that derive from the naturalistic fallacy (*i.e.*, where everything natural is desirable and, therefore, unchangeable) (*Lester, Strunk & Hoover, 2020*; *Lilienfeld et al., 2010*).

The study of misconceptions in Psychology began in the second decade of the 20th century (*e.g.*, *Nixon, 1925*, see also *Tupper & Williams, 1986*). Since then, numerous studies have been carried out, either by reviewing myths in different general areas of Psychology (*e.g.*, *Lilienfeld et al., 2010*) or by focusing on more specific aspects of the discipline, such as myths about schizophrenia (*Furnham & Bower, 1992*), ontological development and neuropsychology (*Furnham, 2018*; *Furnham & Grover, 2019*), intelligence (*Warne, Astle & Hill, 2018*) or behavior analysis (*Lamal, 1995*). However, the study of myths is not limited to behavioral science. The prevalence of myths has been studied in fields such as Education (*Ferrero, Garaizar & Vadillo, 2016*), Medicine (*Kaufman et al., 2013*) or public health policies (*Viehbeck, Petticrew & Cummins, 2015*), as well as in science as a knowledge system (*McComas, 1996*), and the practical consequences that they entail, such as those caused by the myths of mathematics in their students (*Powell & Nelson, 2021*).

## RELEVANT FACTORS IN THE BELIEF IN MYTHS IN PSYCHOLOGY

Several studies show that the progression of the students in psychology education. measured with variables such as the academic year or the course credits earned, is related to the prevalence of myths in the field of Psychology. For example, a positive relationship has been found between academic history and the recognition of myths as such, in addition to a greater predisposition to abandon this type of beliefs (*Bensley, Lilienfeld & Powell, 2014*; *Gardner & Brown, 2013*; *Gardner & Dalsing, 1986*; *Gaze, 2014*). This effect has been found in pre-post (*e.g.*, *Taylor & Kowalski, 2004*; *Taylor & Kowalski, 2012*), cross-sectional (*e.g.*, *Amsel, Baird & Ashley, 2011*; *Gardner & Dalsing, 1986*) and longitudinal design (*McCarthy & Frantz, 2016*).

Some studies suggest that psychology students are better at recognizing the myths in this discipline than the general population or students from other university degrees (*Bensley, Lilienfeld & Powell, 2014*; *Furnham & Hughes, 2014*; *Gardner & Dalsing, 1986*; *Hughes et al., 2015*; *Sibicky, Klein & Embrescia, 2020*). Reaching a higher academic year

could indicate a greater interest in the discipline, and with it, a greater desire to further delve into psychology (*Furnham & Hughes, 2014*). However, although there is a decrease in the belief in myths according to academic year, these preconceptions do not always disappear completely (*Aleksandrova-Howell et al., 2020*; *Brown, 1983*; *Gaze, 2014*). For example, *Gardner & Dalsing (1986)* found that a change only occurs when the person has participated in psychology-related courses to the subject of between 19 and 70 h. In other cases, such as that of Vaughan's study (*1977*), it was found that university students from different degrees started, on average, defending 40% of a series of misconceptions about Psychology. Upon completion of an Introductory Psychology Course at university, participants reduced their misconceptions by only 5.5% (*Vaughan, 1977*). Although none of the studies explicitly state the contents of these introductory courses, which may be a relevant limitation, the results that indicate a relative reduction of these misconceptions seem to be replicated in other studies (*McCarthy & Frantz, 2016*; *Root & Stanley, 2017*). This accumulation of evidence on the decline in myth belief is an indication of the benefits, although limited, that formal education seems to grant as the student progresses through the degree (*Furnham & Hughes, 2014*; *Gardner & Dalsing, 1986*; *Hughes et al., 2015*).

The type of university can also be a relevant factor to consider. The different types of universities can be differentiated, mainly, through their teaching modality: face-to-face or online (*Harmon & Lambrinos, 2006*; *Zhao et al., 2005*). Each teaching modality can attract different types of students. A common practice in the study of myths among the university population has consisted in recruiting students without considering differences according to degree or university in which these higher studies take place (*e.g.*, see *Furnham & Bower, 1992*; *Gardner & Brown, 2013*; *Vaughan, 1977*), to maximize external validity. However, there may be more differences than similarities between students from different university backgrounds. For example, the profile of the average student in a face-to-face public university such as the Autonomous University of Madrid (UAM) is usually that of a young adult between the ages of 18 and 24, who is completing their first undergraduate program, without any added family or work responsibilities (*Universidad Autónomade Madrid, 2019*). In contrast, the profile of the average student from a public online learning university, such as the National Distance Education University (UNED) responds to that of an adult, with full-time work, greater family responsibilities and previous studies, who seeks to improve their professional skills (*García Aretio, 2006*). Regarding age, undergraduate students between 23 and 49 years old tend to choose online universities to pursue an undergraduate program (*Stewart et al., 2010*). Such students tend to work more hours weekly than those who choose face-to-face education (*Dutton, Dutton & Perry, 2002*; *Jenkins & Downs, 2003*; *Mattes, Nanney & Coussans-Read, 2003*) and tend to have children and family responsibilities (*Conklin, 1997*; *Dutton, Dutton & Perry, 2002*; *Grimes & Antworth, 1996*) In this way, each university could contribute a different student profile and, therefore, could show differences in the belief in misconceptions within Psychology. These variations have not been explored in previous studies and could be relevant in the study of myths in Psychology.

Familiarity with scientific dissemination is another factor that may influence the belief in myths. It can be understood as "the use of appropriate skills, media, activities and

dialogue to produce one or more of the following personal responses to science: awareness, enjoyment, interest, opinion-forming and understanding'' of scientific knowledge (*Burns, O'Connor & Stocklmayer, 2003*). There are various formats for its use, but not all of them or their contents are equally adequate. While reading academic literature can help reduce belief in myths (*Furnham, Callahan & Rawles, 2003*), familiarity with content on the Internet can be harmful (*e.g.*, *Putnam, 2011*). This is due, at least in part, to the absence of supervision mechanisms that guarantee that the informative material on the internet transmits information following the scientific evidence available (*Lewandowsky et al., 2012*). Also, non-expert psychologists could have difficulties distinguishing scientific *vs* pseudo-scientific dissemination (see *Lilienfeld, Ammirati & David, 2012*) which could result not only in not reducing the belief in myths, but to increase it. In addition, scientific dissemination can oversimplify scientific facts to make them more accessible to a non-specialized audience, which can lead to misunderstanding research advances and promoting belief in myths (*Lewandowsky et al., 2012*; *Cassany, López & Martí, 2000*). Finally, disseminating false scientific information on social networks can increase its acceptance as correct information simply due to the mere repeated exposure to it, as occurs in other more conventional media (*Begg, Anas & Farinacci, 1992*; *Furnham, Callahan & Rawles, 2003*).

## The methodology used in the assessment of myth belief

To reduce the prevalence of belief in psychological myths, it is first necessary to adequately detect what these myths are, who believes them and how much they believe in them. This is only possible with valid and reliable evaluation methods. However, many studies in this line of research do not report the psychometric properties of their tools, which may affect the generalization of the results when it comes to reliability and validity (*Prieto & Delgado, 2010*). When studying the prevalence of myths and misconceptions in Psychology, self-report questionnaires have been the default choice as they are a set of tools that allow participants to express both their (dis)agreement and the extent to which they believe that certain statements are true (*Bensley & Lilienfeld, 2017*; *Bensley, Lilienfeld & Powell, 2014*; *Hughes, Lyddy & Lambe, 2013*). In this regard, previous literature discusses the relevance of methodological factors such as the response format, content, and psychometric properties of the questionnaires, as well as the characteristics of the samples used in the estimation of myths (*Bensley & Lilienfeld, 2017*; *Hughes, Lyddy & Lambe, 2013*; *Taylor & Kowalski, 2012*). These methodological factors may have influenced how the research was conducted so far, specifically in terms of potential biases (*Taylor & Kowalski, 2012*).

The dichotomous response format (true/false) has been, for many years, the chosen by researchers (*Basterfield et al., 2020*; *Brown, 1983*; *Ferrero, Garaizar & Vadillo, 2016*; *Furnham, Callahan & Rawles, 2003*; *Gardner & Dalsing, 1986*; *Hughes, Lyddy & Lambe, 2013*; *Kowalski & Taylor, 2004*; *Lamal, 1995*; *Taylor & Kowalski, 2004*; *Taylor & Kowalski, 2012*; *Torres, Boccacini & Miller, 2006*; *Vaughan, 1977*). This format is associated with a series of limitations (*Bensley & Lilienfeld, 2017*; *Griggs & Ransdell, 1987*; *Hughes, Lyddy & Lambe, 2013*). One of them is the acquiescence bias, where participants always respond with an affirmative answer (*Bensley & Lilienfeld, 2017*), thus recommending that the

formulation of the statements should be divided into direct and inverse items to prevent response patterns (*Taylor & Kowalski, 2004*; *Griggs & Ransdell, 1987*). If all items represent myths and are worded in such a way that the correct answer is "true", the direct consequence is the overestimation of misconceptions in students as a methodological artifact (*Bensley & Lilienfeld, 2015*; *Griggs & Ransdell, 1987*; *Taylor & Kowalski, 2012*). However, not all studies have carried out this reversal process on items' wording (*e.g.,* see *Gardner & Dalsing, 1986*; *Vaughan, 1977*). When the presentation of the items is balanced (*i.e.,* it is established that approximately half of the items are direct and the other, inverse), the resulting prevalence of believing in myths is lower (*Brown, 1983*).

Another limitation of the T/F format is the impossibility of distinguishing between those responses that represent believing in myths and those that are the product of randomly answering one of the two possible responses (*Taylor & Kowalski, 2004*). One way to control this shortcoming has been to add "I don't know" as a third response option (*Furnham, 2018*; *Furnham, Callahan & Rawles, 2003*; *Gardner & Brown, 2013*; *Gardner & Dalsing, 1986*; *Hughes, Lyddy & Lambe, 2013*; *Lamal, 1995*). Although this can add some bias when evaluating those who are not motivated to answer, its introduction makes it possible to discriminate what the students do not know from what they assume as facts and obtain an answer instead of, perhaps, not answering some items—leaving them blank (*Furnham & Hughes, 2014*; *Furnham, Callahan & Rawles, 2003*). The introduction of 'I don't know' response option comprises around 12–13% of the responses (*Gardner & Brown, 2013*; *Gardner & Dalsing, 1986*) and reduces the estimate of the prevalence of myths by 8%, which generates a closer and more realistic view of the phenomenon (*Gardner & Dalsing, 1986*). However, there are still limitations to be solved (*Hughes, Lyddy & Lambe, 2013*).

As an alternative to dichotomous T/F scales, the use of Likert-type scales is proposed in this type of research (*Gardner & Brown, 2013*; *Gardner & Dalsing, 1986*; *Furnham & Hughes, 2014*). By representing different levels of response, it is possible to discriminate with greater certainty the degree of confidence that the participant has in the veracity of the statements. Several authors (*Furnham & Hughes, 2014*; *Gardner & Brown, 2013*) use questionnaires with scales ranging from "*completely false*" to "*completely true*", allowing for a greater range of answers and not reducing it to a dichotomous choice. Thus, what is being measured is what the participants believe in a more nuanced way than with the dichotomous response format. One problem with this kind of scale is how participants use the central scores. One way is that they use it to report that they do not know if statement is true or false. However, also participants could used it to report that the sentence is half true. In our view, it is important to give participants the option to report that they don't know about the sentence. Because this reason, we labelled the central value of the scale as "don't know". As commented above, that allow a more accurate measure of the participants' beliefs.

Others more critical with the methodology have highlighted the content of the questionnaires. The inclusion of myths that are unrepresentative or irrelevant to the discipline, that are not covered by introductory texts, or those that cannot be identified as such, given the available evidence, can bias, in the same way, the estimation of the belief in

myths (*Bensley & Lilienfeld, 2017*; *Gardner & Dalsing, 1986*; *Griggs & Ransdell, 1987*; *Taylor & Kowalski, 2012*).

Regarding the samples used, the general sample size in this type of study ranges between 100 and 200 participants (*e.g.*, see *Bensley & Lilienfeld, 2015*; *Brown, 1983*; *Furnham & Bower, 1992*; *Gardner & Brown, 2013*; *Taylor & Kowalski, 2012*; *Vaughan, 1977*). In studies that involve the use of questionnaires, the use of a sample size of between 200 and 400 participants is recommended to carry out an adequate psychometric analysis (*Abad et al., 2011*). However, their characteristic, so far, has been their low homogeneity in terms of participants' population (*Bensley & Lilienfeld, 2015*; *Bensley & Lilienfeld, 2017*; *Furnham & Hughes, 2014*). The participants in this type of research are usually university students from different fields of knowledge who enrol in some Introductory Psychology courses (*e.g.*, see *Furnham & Hughes, 2014*; *Gardner & Brown, 2013*; *Kowalski & Taylor, 2009*). This may favor the generalization of the results to the general population, which is a relevant objective, but makes it difficult to study the evaluation of these ideas among psychology students throughout their academic career—and it is relevant to know what happens within formal education. A consequence of this could be the variability presented in the systematic reviews carried out so far. *Hughes, Lyddy & Lambe (2013)* suggest that students who begin Introductory Psychology courses accept between 28% and 71% of these myths. Other studies indicate a recognition that ranges between 30% to 39% of misconceptions (*Taylor & Kowalski, 2012*).

Despite the extensive research on myths in Psychology in other countries—see, for example, Russia (*Aleksandrova-Howell et al., 2020*), North America (*Basterfield et al., 2020*; *Bensley & Lilienfeld, 2017*; *Brown, 1983*; *Gardner & Brown, 2013*), United Kingdom (*Furnham, Callahan & Rawles, 2003*; *Furnham & Hughes, 2014*; *Furnham & Grover, 2019*), or India (*Kishore et al., 2011*)—a study of the same characteristics has not yet been carried out in Spain.

## THE PRESENT STUDY

The present study aims to provide empirical evidence regarding the belief in myths related to Psychology among students enrolled in the Psychology undergraduate program, considering the previously mentioned variables: the academic year, the university in which they are enrolled and the familiarity with scientific dissemination. To do so, an adaptation of one of the available methodological tools with a Likert scale format and with adequate psychometric guarantees is used, in addition to a large sample size to ensure the representativeness of the phenomenon.

## METHOD

### Participants

A sample of 916 students from the Degree in Psychology from two Spanish universities was used: Autonomous University of Madrid (UAM) (face-to-face, $n = 364$, 86% women, aged between 17 and 55 years, $M = 20.03$, $SD = 3.58$) and National Distance Education University (UNED) (online, $n = 552$, 77% women, aged between 18 to 78 years; $M = 35.8$,

*SD* = 11.6) (see Table S1 for an overview of the number of participants by academic year and university).

## Procedure and materials

Data were collected through the *PsInvestiga* participant recruitment system at the UAM ($n = 238$) and dissemination by institutional email in both universities ($n = 678$). All participants completed an online informed consent before the questionnaire.

An online questionnaire was sent to participants . This questionnaire measured the demographic data and the belief in Psychology myths. Participants had to fill out the questionnaire individually and their responses were completely anonymous. Before carrying out the task, the student was informed of its purpose and the voluntary nature of their participation. A *demo* of the questionnaire used can be found at this link: https://forms.gle/6J1EkTT7S9G1iLBH7. The realization of this project, as well as the treatment of the collected data, has been approved by both the Research Ethics Committee of the Autonomous University of Madrid (code CEI-95-1758) and by the Ethics Subcommittee of the Faculty of Psychology from the same university.

### Demographic measures

We collected the following data of each participant: age, gender, any other degree previously obtained, the university in which he/she is currently studying Psychology, the highest year of the degree being studied and his/her familiarity with scientific dissemination. Finally, through an open-ended question, participants specified examples of dissemination sources they consult.

### Myths in psychology

An adaptation of the 55-item questionnaire developed by *Gardner & Brown (2013)* was carried out. The authors received permission to use this instrument from the copyright holders. Five items were eliminated as they were considered to have little relevance or representativeness for the study and 24 new items were added so that the final questionnaire consisted of 74 items (see Table S2 for items excluded and included in the questionnaire). The 74 items, both those used by *Gardner & Brown (2013)* and those added by authors, were Psychology extracted statements taken from the book *50 Great Myths of Popular Psychology* (*Lilienfeld et al., 2010*) accompanied by a five-point Likert-type scale, ranging from 1 (*completely sure it is false*) to 5 (*completely sure it is true*), with 3 as the central category (*I don't know*). To control for response bias, some items were reversed: 55.4% of the items (41) were direct and the remaining 44.6% (33) were reversed. Therefore, some items were formulated directly with the statement that is false according to scientific evidence and, in others, the wording indicated the statement was supported by the available evidence. The content of the test sampled different areas of Psychology in a balanced way. When it was presented in Spanish, the standard procedure was followed for its adaptation to the language: translation from English to Spanish by one of the authors, reverse translation by another co-author -from Spanish to English- and a final check carried out by a native expert (independent from the study) to ensure that the two English versions (the original and the back—translation) were equivalent. A psychometric analysis of this tool revealed

a Cronbach's alpha indicator of 0.85, which indicates acceptable reliability (*Abad et al., 2011*).

## DESIGN AND DATA PROCESSING

In the present study, a cross-sectional descriptive between-subjects design was used, where the dependent variable was the total score of the "Myths in Psychology" scale. This variable is calculated through the sum of all the scores of each participant's responses with the recoded items (those in which such transformation is required). The higher the scores on the questionnaire, the greater the belief in the myths of this discipline. Three independent variables were taken into account: academic year (with four levels: 1, 2, 3, & 4), type of university (with two levels: face-to-face and online) and familiarity with scientific dissemination (with two levels: Yes/No).

To explore the effect of the academic year achieved and of the profile of the Psychology student on the belief in myths, a two-factor ANOVA was performed for independent samples (4 × 2) on the DV "total score" with "year" and "type of university" as factors. On the other hand, a Student's *t*-test was performed to analyze the effect of the familiarity with scientific dissemination on the prevalence of these beliefs. It was not incorporated into the previous analysis of variance because of the limitations associated with the number of participants who indicated familiarity with scientific dissemination and the impact it has on the assumptions of the statistical model.

Following the recommendations on the dissemination of scientific data (*e.g.*, see *Björk et al., 2010*; *Frankenhuis & Nettle, 2018*), the database obtained and used in this study is published on the Open Science Framework platform and is available to any researcher who wishes to consult them at the following link: https://osf.io/tazg9/.

## RESULTS

### Descriptives

The descriptive statistics for the items of the *Myths in Psychology* questionnaire can be found in Table 1, recoded and ordered from highest to lowest score. The graphs of the relative frequencies of items in the total sample can be found in Figs. S1–S4. The mean of the assessment at item-level across the total sample was 2.62 ($SD = 0.16$). If the data were differentiated according to the type of university, in the face-to-face one there was a value of 2.39 ($SD = 0.82$) and, in the online one 2.79 ($SD = 0.15$).

As mentioned, each participant could evaluate each item on a 1–5 scale. We identified means around 4 and 5 [>3.5] (*I think it is true* and *Completely sure it is true*) as misconceptions that are accepted by participants. In the other hand, means around 1 and 2 [<2.5] (*Completely sure it is false* and *I think it is false*) are considered as detected myths. Therefore, the total sample believed in a total of 9 misconceptions (12.16%); the participants from face-to-face university in 10 (13.51%) and the students of online, in 13 (17.56%). Most of the responses revolved around the central score, "I don't know" [2.5–3.5]: for 30 items (40.54%) in the total sample, for 22 (29.73%) in face-to-face and for 38 (51.36%) in online. On the other hand, in the total sample, a total of 35 myths (47.3%)
**Table 1  Means and standard deviations (in parentheses) of the questionnaire items as a function of the total and separated by university.**
Ordered from highest to lowest by the score obtained in the total sample. Inverse test items are included, recoded, so that a higher mean indicates greater belief in the misconception.

| ITEM | Total Mean (SD) | Face-to-face Mean (SD) | Online Mean (SD) |
|---|---|---|---|
| 13. Individuals commonly DON'T repress the memories of traumatic experiences. (*) | **4.27 (1.04)** | **4.03 (1.14)** | **4.44 (0.94)** |
| 25. Students learn best when teaching styles are matched to their learning styles. | **4.22 (0.9)** | **4.10 (0.88)** | **4.30 (0.90)** |
| 12. When dying, people DON'T pass through a universal series of psychological stages. (*) | **4.12 (1.07)** | **3.83 (1.13)** | **4.32 (0.98)** |
| 5. Subliminal messages can't persuade people to perform some behaviors. (*) | **4.1 (1.2)** | **3.95 (1.23)** | **4.29 (1.16)** |
| 17. Some people have true photographic memories. | **4.07 (1.02)** | **3.75 (1.12)** | **4.29 (0.89)** |
| 58. Most people who experience severe trauma, (e.g., as in military combat) DON'T develop posttraumatic stress disorder (PTSD). (*) | **3.96 (1.07)** | **3.80 (1.11)** | **4.07 (1.04)** |
| 68. More experienced therapists are generally NO more effective than those with little experience. (*) | **3.77 (1.09)** | **3.76 (1.06)** | **3.77 (1.11)** |
| 28. Hypnotized people are aware of their surroundings and can recall the details of conversations overheard during hypnosis. (*) | **3.7 (1.12)** | *3.32 (1.15)* | *2.90 (1.07)* |
| 72. Electroconvulsive therapy is rarely administered today. | **3.67 (1.10)** | **3.7 (1.09)** | **3.66 (1.12)** |
| 26. Direct instruction is superior to discovery learning. (*) | *3.49 (1.2)* | *3.41 (1.14)* | **3.54 (1.24)** |
| 46. People's attitudes are NOT highly predictive of their behaviors. (*) | *3.48 (1.21)* | *3.25 (1.26)* | **3.62 (1.16)** |
| 21. There is a modest correlation between brain size and IQ in humans. (*) | *3.44 (1.24)* | **3.55 (1.31)** | *3.36 (1.18)* |
| 67. Crowding consistently leads to more aggression. | *3.43 (1.08)* | **3.61 (1.03)** | *3.30 (1.10)* |
| 38. Positive thinking is better than negative thinking for all people. | *3.38 (1.31)* | *2.95 (1.28)* | **3.67 (1.25)** |
| 15. The memory of everything we've experienced is stored permanently in our brains, even if we can't access all of it. | *3.36 (1.39)* | *2.91 (1.42)* | **3.66 (1.30)** |
| 64. Most people that plead insanity are NOT faking mental illness. (*) | *3.32 (1.04)* | *3.08 (0.95)* | *3.47 (1.07)* |
| 33. Awakening a sleepwalker is NOT dangerous. (*) | *3.29 (1.4)* | *3.16 (1.42)* | *3.37 (1.36)* |
| 7. People become increasingly satisfied with their lives in old age. (*) | *3.20 (1.09)* | *3.22 (1.08)* | *3.18 (1.10)* |
| 50. Most children survive the divorce of their parents without much, if any, long-term psychological damage. (*) | *3.20 (1.22)* | *2.88 (1.23)* | *3.42 (1.17)* |
| 35. Ulcers are caused primarily by stress. | *3.15 (1.1)* | *3.08 (1.04)* | *3.20 (1.13)* |
| 39. Voice stress analyzers can help to detect lying. | *3.13 (1.23)* | *2.53 (1.21)* | *3.52 (1.08)* |
| 23. Irregularly provided feedback best promotes long-term learning. (*) | *2.98 (1.25)* | *3.02 (1.27)* | *2.95 (1.25)* |
| 60. Hallucinations are almost always a sign of serious mental illness. | *2.93 (1.29)* | *2.54 (1.20)* | *3.19 (1.28)* |

were recognized as such; with 42 (56.76%) at face-to-face, and 23 (31.08%) at online (see Table 1).

The range of total scores in the questionnaire goes from 74 (score 1 in all myths) to 370 (score 5 in all myths). The higher the score, the most belief in myths. For the total sample was obtained a mean score of 194.82 points (*SD* = 25.11, *min.* = 128, *max.* =
**Table 1** (*continued*)

| | Total | Face-to-face | Online |
|---|---|---|---|
| ITEM | Mean (SD) | Mean (SD) | Mean (SD) |
| 31. Our brains rest during sleep. | 2.91 (1.5) | 2.57 (1.41) | 3.13 (1.52) |
| 36. Women are NO better than men at accurately guessing the feelings of others. (*) | 2.91 (1.34) | 2.4 (1.28) | 3.26 (1.28) |
| 4. Extrasensory perception is NOT a real phenomenon. (*) | 2.88 (1.42) | 2.38 (1.43) | 3.22 (1.32) |
| 44. The best way to change someone's attitude is to give them a large reward to do so. | 2.83 (1.31) | 3.08 (1.34) | 2.66 (1.27) |
| 42. Expressing anger directly toward another person or object makes us more aggressive. (*) | 2.81 (1.17) | 3.03 (1.07) | 2.67 (1.20) |
| 8. Most adopted children are psychologically healthy. | 2.8 (0.99) | 2.99 (1.02) | 2.71 (0.96) |
| 49. Most people who were physically abused as children DON'T go on to become abusers themselves. (*) | 2.8 (1.21) | 2.53 (1.22) | 2.98 (1.18) |
| 57. All clinically depressed people suffer from extreme sadness. | 2.78 (1.32) | 2.62 (1.30) | 2.89 (1.33) |
| 20. IQ scores are relatively unstable during childhood. (*) | 2.77 (1.08) | 3.04 (1.10) | 2.59 (1.03) |
| 61. Homicide is more common than suicide. | 2.73 (1.24) | 2.72 (1.22) | 2.74 (1.26) |
| 22. As a general rule. students typically recall only 10% of what they read. | 2.71 (1.1) | 2.35 (1.04) | 2.95 (1.07) |
| 71. Taking a placebo (i.e., sugar pill) can change brain functioning and its chemistry. (*) | 2.68 (1.38) | 2.66 (1.39) | 2.70 (1.37) |
| 24. Negative reinforcement is a type of punishment. | 2.65 (1.61) | 1.88 (1.37) | 3.16 (1.55) |
| 55. People with schizophrenia DON'T have multiple personalities. (*) | 2.65 (1.3) | 2.31 (1.23) | 2.88 (1.30) |
| 11. Playing classic music to infants DON'T boosts their intelligence. (*) | 2.63 (1.4) | 1.82 (1.19) | 3.16 (1.27) |
| 29. It is impossible to lie under hypnosis. | 2.52 (1.01) | 2.34 (1.03) | 2.64 (0.98) |
| 27. Hearing material while we are asleep (sleep learning) can be an effective aid to learning. | 2.42 (1.25) | 1.8 (1.07) | 2.84 (1.20) |
| 63. The words ''insanity'' and ''sanity'' are purely legal NOT psychological terms. (*) | 2.4 (1.16) | 2.23 (1.08) | 2.52 (1.19) |
| 73. Expert judgment and intuition are the best means of making clinical decisions. | 2.39 (1.22) | 2.11 (1.16) | 2.58 (1.24) |
| 37. Unfamiliarity breeds contempt: We dislike things we have less exposure to. (*) | 2.37 (1.12) | 2.19 (1.09) | 2.50 (1.13) |
| 45. Men and women communicate in completely different ways. | 2.3 (1.18) | 1.81 (0.98) | 2.61 (1.19) |
| 53. People's responses to inkblots tell us a great deal about their personalities. | 2.29 (1.29) | 1.52 (0.97) | 2.80 (1.22) |
| 43. Groups tend to make less extreme decisions than individuals. | 2.28 (1.4) | 1.94 (1.23) | 2.52 (1.45) |
| 9. Married couples enjoy more marital satisfaction after they have children. | 2.23 (0.94) | 2.29 (0.95) | 2.18 (0.94) |
| 18. Human memory works like a tape recorder, and accurate records events we've experienced. | 2.16 (1.23) | 1.54 (0.94) | 2.57 (1.33) |
| 40. We are most romantically attracted to people who are similar to us. (*) | 2.15 (1.15) | 1.93 (1.11) | 2.30 (1.15) |

**Table 1** (*continued*)

| ITEM | Total<br>Mean (SD) | Face-to-face<br>Mean (SD) | Online<br>Mean (SD) |
|---|---|---|---|
| 74. Most modern therapies are NOT based on the teachings of Freud. (*) | 2.14 (1.14) | 1.92 (1.06) | *2.29 (1.17)* |
| 59. Psychiatric hospital admissions and crimes DON'T increase during full moons. (*) | 2.12 (1.39) | 1.46 (1.07) | *2.56 (1.41)* |
| 56. There has recently been a massive epidemic of childhood autism. | 2.08 (1.10) | 2.03 (1.09) | 2.11 (1.12) |
| 1. Most people use only about 10% of their brain power. | 2.07 (1.38) | 1.24 (0.76) | *2.62 (1.43)* |
| 6. Humans have an invisible body energy that can cause psychological problems when blocked. | 2.04 (1.25) | 1.44 (0.75) | 2.44 (1.36) |
| 62. Most rapes are committed by strangers. | 1.99 (1.02) | 1.96 (1.04) | 2.02 (1.02) |
| 19. Rote memorization is NOT the best way to retain information. (*) | 1.97 (1.13) | 1.86 (1.05) | 2.05 (1.18) |
| 34. The polygraph (lie detector) test is NOT an accurate means of detecting dishonesty. (*) | 1.96 (1.16) | 1.36 (0.81) | 2.35 (1.19) |
| 66. Rehabilitation programs have NO effect on the recidivism rates of criminals. | 1.96 (0.96) | 1.87 (0.87) | 2.02 (1.02) |
| 41. The more people present at an emergency, the greater the chance that someone will intervene. | 1.95 (1.34) | 1.84 (1.35) | 2.03 (1.32) |
| 14. People with amnesia can still recall some details of their earlier lives. (*) | 1.88 (0.97) | 1.78 (0.93) | 1.95 (1.00) |
| 16. With effort, we can remember events back to the time of our birth. | 1.86 (1.05) | 1.46 (0.72) | 2.13 (1.16) |
| 2. Almost all color-blind people can see at least some colors. (*) | 1.84 (1.06) | 1.85 (1.026) | 1.84 (1.08) |
| 47. We CANNOT tell a person's personality by merely looking at their handwriting. (*) | 1.84 (1.19) | 1.23 (0.72) | 2.24 (1.26) |
| 52. The fact that a trait is heritable means we CAN'T change it. | 1.83 (1.1) | 1.64 (1.02) | 1.96 (1.14) |
| 69. Most psychotherapy involves a couch and exploring one's early past. | 1.8 (1.15) | 1.66 (1.11) | 1.89 (1.17) |
| 51. Obese people are more cheerful ("jolly") than thin people. | 1.71 (0.92) | 1.6 (0.82) | 1.87 (0.96) |
| 65. Most psychopaths are violent. | 1.68 (0.88) | 1.52 (0.77) | 1.80 (0.93) |
| 3. Some people are exclusively left-brained while others are right-brained. | 1.67 (1.01) | 1.27 (0.55) | 1.94 (1.16) |
| 54. Only deeply depressed people commit suicide. | 1.64 (0.97) | 1.47 (0.9) | 1.75 (1.00) |
| 70. Antidepressants are much more effective than psychotherapy for treating depression. | 1.63 (0.85) | 1.54 (0.81) | 1.70 (0.87) |
| 30. Virtually all people dream. (*) | 1.6 (0.96) | 1.64 (0.93) | 1.58 (0.98) |
| 10. Infants establish attachment bonds only to their mothers. | 1.45 (0.87) | 1.12 (0.44) | 1.66 (1.01) |
| 48. Knowing a person's astrological sign predicts their personality traits at better than chance levels. | 1.42 (0.84) | 1.18 (0.61) | 1.59 (0.94) |
| 32. Researchers have demonstrated that dreams possess symbolic meaning. | 1.36 (1.3) | 1.77 (1.09) | *2.75 (1.29)* |

**Notes.**

Items in bold exceed the "myth" threshold established by the researchers (Mean > 3.5). Items in italics are included in the "I don't know" category (2.5 ≥ Mean ≥ 3.5).

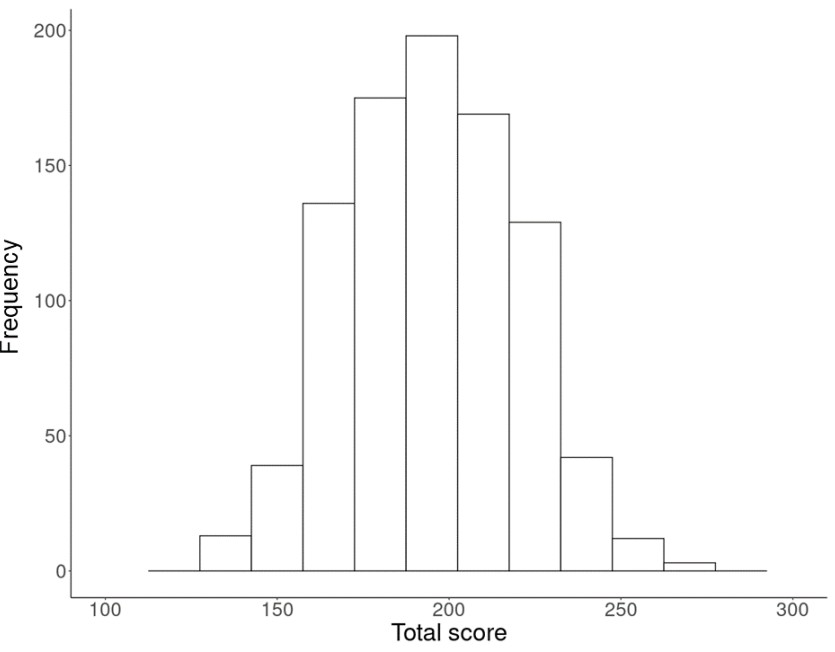

**Figure 1  Histogram of the total score obtained in the questionnaire.**

275) (Fig. 1). If the information was analyzed according to the type of university, online presented a higher score ($M = 206.391$, $SD = 22.15$, *min.* $= 138$, *max.* $= 275$), in relation to that obtained in face-to-face ($M = 177.274$, $SD = 18.25$, *min.* $= 128$, *max.* $= 240$). Higher scores were found among first year students ($M = 202.16$; $SD = 24.84$) compared to those in the final year of the Degree ($M = 181.55$; $SD = 20.95$) (see Table S3 to consult the means of the total score, according to university and the academic year).

## Effects of type of university and academic year on belief in myths in psychology

There was a statistically significant main effect of the type of university factor associated with a large effect size, following Cohen's criteria (*1988*), $F(1,908) = 65.908$, $p < 0.01$, $\eta_G^2 = 0.216$. This effect shows that face-to-face students believe to a lesser extent in this kind of myths, as the mean total score of this university was lower ($M = 177.27$) than that found among the online students ($M = 206.39$).

Although with small effect size, a statistically significant effect was found for the academic year variable, which indicates that beliefs differ according to the year in which the participants are enrolled $F(3,908) = 27.256$, $p < 0.01$, $\eta_G^2 = 0.083$. Multiple *post hoc* comparisons with the Bonferroni correction indicated significant differences between the first and the rest of the academic years ($p < 0.01$), as well as between the second and the first year and between the second and the fourth year ($p < 0.01$) (see Fig. 2).

There was a statistically significant effect of the interaction between these two variables. The influence of the academic year on belief in myths was different for students from the two universities participating in the sample, participants from the face-to-face university

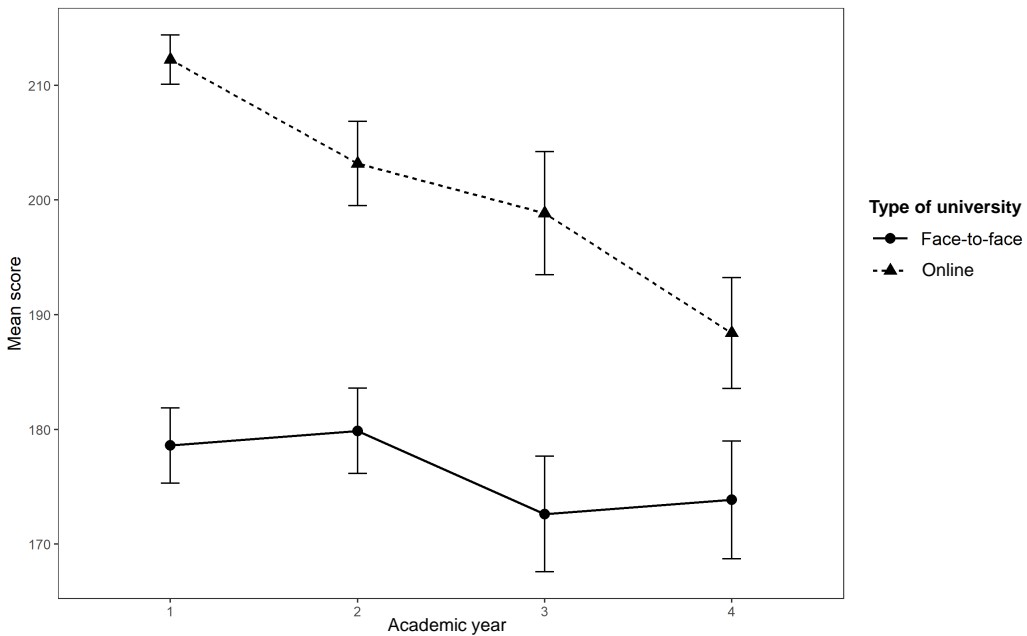

**Figure 2  Effect of the interaction between university and academic year on the total score of the questionnaire.**

were less prone than online one to believe in myths, while the latter benefited more from formal education. However, the effect size found was small ($F$ (3,908) $= 8.944$, $p < 0.01$, $\eta_G^2 = 0.029$) (Fig. 2). The summary of the ANOVA results can be found in Table S4.

### Effect of familiarity with scientific dissemination

A total of 23.62% ($n = 86$) of the face-to-face university participants claimed familiarity with scientific dissemination. At the online, this figure rose to 27.17% ($n = 150$). In total, 236 participants (25.76%) reported performing this activity (see Table S5 for a scientific dissemination sources classification).

The performance of a Student's *t-test* for independent samples with an associated statistic $t_{425,05} = 3,301$ ($p < 0.001$; $IC95\% = [2.470, 9.742]$) indicates that there was a statistically significant effect of familiarity with scientific dissemination compared to non-familiarity with the belief in myths of Psychology. This effect favors those who claim to do so, by obtaining a lower mean in the total score (Fig. 3 and Table S6). However, the effect size found was small (Cohen's $d = 0.244$).

## DISCUSSION

The present study aimed to analyze the prevalence of psychology-related myths in Psychology students in Spain and the influence of three variables on this prevalence: the academic year they are enrolled in, the university in which they study (associated with different characteristic student profiles) and their familiarity with scientific dissemination. The data analyzed sustained that, within the sample used, the variables of academic year,

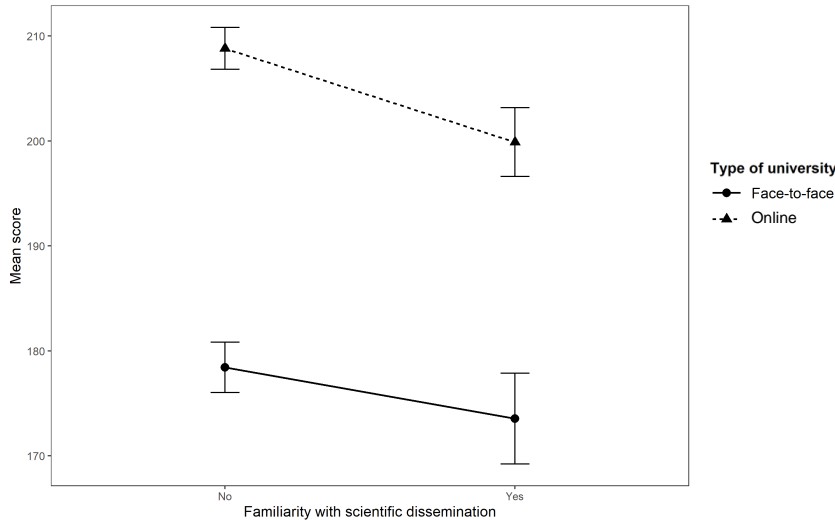

**Figure 3** **Effect of the familiarity with scientific dissemination on the total score (by university).**

type of university and familiarity with scientific dissemination had a statistically significant effect on the belief in myths in Psychology. The chosen sample size favored the external validity of the study and the generalization of its results to similar populations. According to the data we obtained, it could be concluded that students were capable of recognizing most of the myths that appeared in the questionnaire as such, while they still lacked certain limited knowledge regarding certain aspects of the discipline. Thereby, belief in misconceptions regarding Psychology in the chosen sample cannot be considered high, compared to the values indicated in *Hughes, Lyddy & Lambe's (2013)* review or compared to the original study by *Gardner & Brown (2013)*.

The effects found concerning to the academic year within the Degree in Psychology and the familiarity with scientific dissemination were statistically significant. However, the reported effect sizes were small. This refers to a limitation of the benefits of the different educational tools (formal, as is the case of the degree; informal, in the case of scientific dissemination), which is consistent with the previous literature (*e.g.*, see *Furnham & Hughes, 2014*; *Hughes, Lyddy & Lambe, 2013*). Likewise, the trend that was already reflected in the scientific literature was noted - that as the academic year progressed, the proportion of myths decreased (*e.g.*, *Gardner & Dalsing, 1986*).

The effect of the type of university was also statistically significant, with an associated effect size larger than that of the other two factors. Similarly, the academic year had a differential effect on the belief in these myths depending on the university where the psychology studies were carried out. Thus, the reduction in the prevalence of myths that was perceived through the academic year was more prominent at online university. Perhaps it is because these students started from a much higher level of misconceptions. This may be due to the different demographic characteristics that separate these students into different groups as, for example, age. While the face-to-face university, UAM, sample was composed of young people studying their first university degree, the online, UNED, sample showed a

mean age difference of one decade. Also, participants from online use to have qualitatively higher familiar and working responsibilities compared to face-to-face participants. City or country of residence could also affect to these results, whereas face-to-face students are living in Madrid or near cities, the online ones are spread all over the world. However, we are aware that our data by themselves cannot explain these differences and more research must be conducted to study them and to find out their causes.

Previous studies suggest that, in general, the higher the level of studies, the less belief in myths (*Furnham & Hughes, 2014*; *Gardner & Dalsing, 1986*). With these data, it could be postulated that online students should believe less in myths. However, these studies have been carried out in comparison with the general population, and not within the Psychology student body itself. Having accumulated prior knowledge of different areas or the priority that the studies they are studying have in their day to day can be examples of other variables that are influencing the belief of myths, ensuring a higher score on this type of scales.

It is necessary to emphasize that the effect of formal education that we found is quite limited. Possible explanations for this phenomenon would lie in the analysis of what happens in the classroom. The refutational techniques on popularized ideas about Psychology have proven to be effective in this case; however, they are underused or absent (*Bensley & Lilienfeld, 2015*; *Gardner & Dalsing, 1986*). The illustration that is made in some subjects of Psychology as a set of isolated plots of knowledge, without any relation to each other, can also be counterproductive. This is consistent with what some studies pointed out: after discussing misconceptions in class, belief in them decreases, but re-emerges when moving to a different topic (*Lyddy & Hughes, 2012*). It is the so-called *rebound effect* (*Lewandowsky et al., 2012*). That is, the explanations that are given do not acquire the degree of generalization necessary for them to produce stable modifications over time in the beliefs of the students. In addition, it is possible that belief in myths could be enhanced within the universities. Some authors indicate that problems when teaching introductory courses on the philosophy of science that supports Psychology can influence the subsequent development of misconceptions in this regard (*García García et al., 2006*). By not knowing the scope, implications, and characteristics of this discipline, it is difficult to distinguish between what is valid and what is not.

In this study, familiarity with scientific dissemination proved to have a positive but limited effect. Only a small proportion of the sample claimed to carry out such activity and it was more frequent among the online participants. Two characteristics shared by some sources of dissemination considered is that they are accessible and practical, which favors their propensity. The journals that have the title "Psychology" and other complementary words, widely disseminated without their recognition being due to the scientific quality of their contents, may present partial truths or biased statements that, although well directed, are not completely correct. As they are not reviewed by accredited specialists in the field, there is no possible filter. Part of the reduced effect size found could be explained by this. As reported in *Lewandowsky et al. (2012)*, the fact that now people stopped having a passive role and have begun to create content in the media (often online) can spread misinformation, rather than having an exclusively positive side. Also, social media allow the possibility of creating a community between users with the same interests is added.

While the debates that take place online can be enriching, the effect that can be obtained is the opposite when handling information that is not verified. Some studies have already explored the possible advantages of using social networks such as Twitter in the academic world (*e.g.*, see *Letierce et al., 2010*). Students must be provided with the appropriate tools that will allow them to make the most of these information options.

The present study has also some limitations to consider. Firstly, the item presentation order was predefined by experimenters and the same for all participants. This could cause some undesirable effects such as fatigue or practice effects, that affect to participants' responses. Future studies may consider to randomize or counterbalance the order of item presentation to avoid these effects and their influence on the measure of the believe in myths. Also, future studies could measure, not only the belief in myths, but also the participants' knowledge of true ideas about human behavior. That will allow researchers to compare these two approaches and explore if there is a relation between them.

Another limitation of our study is related with the measure of the scientific dissemination variable, what could negatively affect our results and make them less precises. While the university and the academic year are easy to answer questions, measuring the familiarity with scientific dissemination is less so. For example, participants may don't know what scientific dissemination is or have an incorrect idea of it. Also, participants could be not able to distinguish scientific *vs* pseudo-scientific dissemination, which not only will decrease the belief in myths but could it makes increase. Finally, we did not distinguish how much participants are familiar with scientific dissemination. Although our results pointed out that those participants that reported scientific dissemination believing less in myths than those that do not in both universities, to assess this factor more precisely and to delve deeper into how these two variables—being familiar with scientific dissemination and how much so- are related, probably using more than one question to do so, to obtain a more sensible measure.

As far we know, the authors of the original test we used did not analyze its construct and convergent validity, neither we do. In order to provide more reliable information, this could be analyzed and reported. Finally, it is recommendable to estimate the sample size needed to achieve adequate statistical power. Here we did not estimate it, but we tried to obtain the most participants as possible. Future research could calculate the minimun necessary sample size following the recommendations for this kind of research (*e.g.*, *Hair et al., 2010*).

## CONCLUSIONS

The results of the present study offer a sample of the belief in myths in Psychology among students of this discipline in Spain. Thus, there is no clear defense of myths as true statements by psychology students, although a large number of responses were found to be around the mean value of the scale ("*I don't know*"). The main conclusions of the study could be summarized in these three: (a) Students enrolled in higher academic years tend to believe less in myths than those who coursing initial years; (b) Familiarity with scientific dissemination is paired to a decrement in the belief in myths, although its benefits seem

restricted in scope; (c) Prevalence of belief in myths varies according to the characteristics of the universities of origin, in such a way that students enrolled in a online university belief more than those who belong to a face-to-face one. Another interesting finding is that participants of online university experience a greater reduction in belief in this type of misconceptions through year, but they do not reach the level from which students enrolled in face-to-face universities start.

It is known that accepting myths in psychology as true correlates with accepting other misconceptions in other fields, such as the belief in paranormal events or in pseudoscientific practices as true practices (*Bensley, Lilienfeld & Powell, 2014*). Given that this type of study is beginning to be carried out in Spain, future lines of research should explore other possible variables, beyond those analyzed in this study, for their connection with the belief in these types of ideas. Identifying the myths currently held by students is the first step in establishing effective interventions. Treating these misconceptions early is necessary to train professionals who, on the applied side, provide the most effective services for their users and that, on the theoretical side, do not compromise the advancement of behavioral science with these obstacles.

## ACKNOWLEDGEMENTS

Special thanks to Miguel A. Vadillo and Marta Ferrero for their participation in the research design, the adaptation of the questionnaire used, and for making the work so enjoyable.

### Funding
The authors received no funding for this work.

### Competing Interests
The authors declare there are no competing interests.

### Author Contributions
- Cristina Rodríguez-Prada conceived and designed the experiments, performed the experiments, analyzed the data, prepared figures and/or tables, authored or reviewed drafts of the article, and approved the final draft.
- Cristina Orgaz conceived and designed the experiments, performed the experiments, authored or reviewed drafts of the article, funding, and approved the final draft.
- Carmelo P. Cubillas conceived and designed the experiments, performed the experiments, analyzed the data, prepared figures and/or tables, authored or reviewed drafts of the article, and approved the final draft.

### Human Ethics
The following information was supplied relating to ethical approvals ({i.e.}, approving body and any reference numbers):

The Research Ethics Committee of the Autonomous University of Madrid approved the study (CEI-95-1758).

## Data Availability

The raw data, figures and supplementary material are available at OSF: Rodríguez-Prada, Cristina, Carmelo P. Cubillas, and Cristina Orgaz. 2021. "Myths in Psychology: Measurement of Misconceptions among Psychology Students in Spain." OSF. November 15. doi: http://dx.doi.org/10.17605/OSF.IO/TAZG9.

## Supplemental Information

Supplemental information for this article can be found online at http://dx.doi.org/10.7717/peerj.13811#supplemental-information.

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
