# Peer review of "Myths in psychology: psychological misconceptions among Spanish psychology students"

_PeerJ, doi:10.7717/peerj.13811_

## Round 0.1 · original submission · Major Revisions

I have received critiques from two experts. Both are positive about the manuscript, but Reviewer 1 has a series of reasonable concerns that you will need to address in a revision. I don't believe that any of the critiques will provide a substantial barrier to future publication.

I have two minor pieces of feedback to add:
- I have a hunch there is a missing decimal point in the F ratio reported on line 332.
- line 477: "y" should be &

Reviewer 1 ·

Basic reporting

While I think the core premise of the study is interesting- comparison of myths in psychology majors from different types of university settings and in a country that has not been previously investigated but these central contributions are not as clearly stated and addressed.
For comparison of university types-Throughout the paper this should be referred to by their features not the specific school name (online vs traditional). In results the range of ages are reported and seem to overlap similarly so the argument that this is a different population is not as clear as suggested in the introduction.
I find the concept of scientific dissemination a confusing construct and only measured by one yes/no item that it is not clear students understand. The issue is not scientific dissemination itself but how accurate that dissemination is and being able to critically evaluate the source. It is essential that science be disseminated to a larger audience than just scientific journals.
It is only mentioned briefly and not with regard to specific studies but important to discuss the countries in which other data has been collected when making the case to transfer to Spain- clarify if this has been looked at non-US samples previously and how that data compares (you have references from London among others but don’t address this explicitly).

There is awkward wording throughout sections of the paper.
For example, the first sentence of the introduction is confusing and does not translate well into English. The first sentence of the abstract is much clearer and related to the topic you are addressing.
Line 41- the word “proven” is not substantiated
Line 44-46 confusing sentence
Line 58-59 clinical areas would include therapies without empirical evidence
Line 67- need a secondary citation for work in 1920s
Line 92-93 awkward wording of sentence
95-97 range cited is very wide (and is there really a cap to the hours?)
Statistical tests and numbers should not be included in discussion

Experimental design

See note above about clarity of research question and concern about meaurement of "scientific dissemination"
Measures- what criteria were used to eliminate or add items from the existing measure? Mention “irrelevant” but is that based on numbers reported in prior studies?
Raw data supplied. Question about last column (in Spanish) appears to list careers/majors but that does not seem to match the way the sample was described as psychology students (unless it is career goals for seniors?)- no column heading
Study is not "observational" (line 287)

Validity of the findings

When reporting a difference between means, need to include the test that supports that statement (line 313 and 325, 363 as example)
line 343 needs to be new paragraph
Clarify more in comparison to other research impact of "don't know" and whether or not that is included in prior studies (some have, some do not)

·

Basic reporting

Thank you for allowing me to participate in the review process of this study.
The present study analyses the prevalence of false beliefs or psychological myths in Psychology undergraduate students at two Spanish universities. In addition, it analyses how variables such as academic year or familiarity with scientific knowledge media could modulate the prevalence of these false beliefs.
I believe that the study addresses a very important and topical issue. Myths or false beliefs about academic or professional content are a problem in the scientific/academic advancement of scientific areas, so knowing the prevalence of these false beliefs and the variables that could help us to combat them is an interesting contribution to the scientific community.
The paper shows a well-organised theoretical background, and I believe that it helps both the novice and the more experienced reader to contextualise and justify the present study. It also clearly addresses potential methodological problems. The analysis of biases in the use of the tool is very illuminating.

Experimental design

See attachment

Validity of the findings

See attachment

Additional comments

See attachment

---

## Round 0.2 · Major Revisions

Both of the reviewers from round one have assessed your revision. Reviewer 1 has lingering concerns that should be addressed. You will see that they are still concerned about your measure of scientific dissemination. They have suggested a re-framing of the variable that seems quite sensible to me.

Their additional suggestions are all in service of enhancing the clarity of your manuscript.

Reviewer 1 ·

Basic reporting

I still find the concept of scientific dissemination difficult- especially as measured here by the single item. While the letter addresses the concept, I did not see that change in the paper and found the examples confusing- especially as it appears item specifically states not a search in database for academic paper and that is the most frequently reported? (line 341 results)

I find the label "not face-to-face" confusing for school and think either online or distance learning would be more helpful distinction.

In abstract and at times in paper, phrasing sounds longitudinal but comparing different students in different years (courses is awkward wording across countries)

In respsonse to feedback acknowledged international component but not brought into intro

Experimental design

`See comments about scientific dissemination above and in addition may be more helpful to conceptualize as scientific vs pseudo-scientific sources. In US many school incorporate information literacy directly which should give skills to students in general before getting to PSY couses.

Write out school names in Method (if using abbreviations). Why is the % in each grade reported across sample when all analyses are reported by school?

says computerized for design- assume entire study was online
Did IRB at 2nd institution not review? line 241-242
Demographic measures lines 243-249 not complete sentences and dissemination info confusing here

Tell us how many myths you eliminated (line 253), how many news ones added. line 258 are the items from the textbook the original list or the ones you added (or both)

Validity of the findings

Reference to tables and figures is confusing. Primary data should not be in supplemental materials. And why is first table referred to Table 6? line 296. Vague references to Figures 1-3.

lines 309-316 talk about total scores- start there, then average scores.
If not showing full list of myths- should at least identify the 10 and 13 misconceptions- do these overlap or different items (line303-304)
pp 330-334- explain which way the difference went

Additional comments

Also APA style errors for in text citations- when use all authors or how to refer to a study in the wording of a sentence

Annotated reviews are not available for download in order to protect the identity of reviewers who chose to remain anonymous.

·

Basic reporting

Dear Authors,

Thank you very much for giving me the opportunity to review the progress of the manuscript entitled "Myths in Psychology: Psychological misconceptions among Spanish psychology students".
It is very gratifying to see how both the feedback provided by the authors and the changes made to the new version of the manuscript, in my opinion, positively affect the quality of the review process and the resulting manuscript. For this I congratulate the authors.
With regard to the issues raised in my previous review, I consider that the authors have responded satisfactorily to each of them.
I understand that some of the issues I raise go beyond the stated objectives and I am pleased that this has been considered positively by the authors as a basis for further study.
For my part, I consider the manuscript to be of sufficient quality to be published in its current format and I consider it to be a relevant contribution to the literature.
Best regards,
Samuel P. León.

Experimental design

see "Basic reporting"

Validity of the findings

see "Basic reporting"

Additional comments

see "Basic reporting"

---

## Round 0.3 · accepted · Accept

I am satisfied that you have responded to the reviewers' critiques. There are still a few lingering grammatical errors, but all are quite minor. One thing: During typesetting, please be sure to fix the spelling of "Frequency" on the y-axis of Figure 1.

Congratulations!
Tony Barnhart